# Conceptualising Drivers of Illegal Hunting by Local Hunters Living in or Adjacent to African Protected Areas: A Scoping Review

**Paul Zyambo [1],\*, Felix K. Kalaba [2], Vincent R. Nyirenda [3] and Jacob Mwitwa [4]**

1   School of Postgraduate, University of Lusaka, Lusaka P.O. Box 36711, Zambia
2   Department of Plant and Environmental Sciences, School of Natural Resources, The Copperbelt University, Kitwe P.O. Box 21692, Zambia
3   Department of Zoology and Aquatic Sciences, School of Natural Resources, The Copperbelt University, Kitwe P.O. Box 21692, Zambia
4   School of Applied Sciences, Kapasa Makasa University, Chinsali P.O. Box 480195, Zambia
\*   Correspondence: paulzya@yahoo.com; Tel.: +260-978-290-175

**Abstract:** Illegal hunting of wildlife by community members abutting African protected areas contributes to unsustainable use of wildlife, resulting in significant declines in wildlife populations. Contemporary intervention measures have largely been ineffective, leading to pervasive and persistent illegal hunting. Such illegal hunting of wildlife is partly exacerbated by poor understanding of what motivates people to hunt illegally. Applying a scoping review approach, this study aims at developing concepts for drivers of illegal hunting and how they influence illegal hunting behaviour by local hunters living in or adjacent to African protected areas. A total of 30 publications were included for review analysis from 1014 publications retrieved using data base searches on Google Scholar and ScienceDirect. The study identified 12 proximate and five underlying drivers, which were categorised into 10 thematic drivers of illegal hunting by local hunters. The need for survival and sustaining livelihoods was conceptualised as the key thematic driver of illegal hunting by local hunters. The study represents a novel work of conceptualising drivers of illegal hunting by local hunters with implications on the persistence of illegal hunting in Africa.

**Keywords:** Africa; drivers of illegal hunting; illegal hunters' behaviour; local hunters; survival and sustaining livelihoods; wildlife

## 1. Introduction

Illegal hunting of wildlife is prevalent in Africa and has reached crisis levels, as wildlife populations are decimated in 52% of forests, 62% of wilderness areas and 20% of protected areas, thereby threatening sustainability in biodiversity conservation and community livelihoods [1,2]. Illegal hunting refers to any capturing, shooting, killing or extraction of wildlife that is not explicitly sanctioned by the state or private owner of wildlife [3–5], and has possibly persisted in Africa because intervention measures or responses to illegal hunting have been less effective [6]. The sustained illegal hunting is attributed to poor understanding of illegal hunting and what motivates people to hunt illegally [6] and emanates mainly from inadequate empirical information on illegal hunting, a narrow view that it is only a conservation matter and the assertion that it is mainly driven by poverty [5]. However, some evidence does not support these assertions. Duffy et al. [5] indicated that the perspectives on illegal hunting were framed by certain understandings of poverty and that motivations for illegal hunting, such as those arising from complex historical context in regard to the outlaw of community, have not been adequately understood. Travers et al. [7] also found that a lack of alternative employment choices might be a more significant driver for hunters in Uganda than material poverty, which is contrary to the

narrative that people hunt illegally because of poverty. Thus, the narratives on illegal hunting and its drivers may have been inadequate and simplistic. It is for this reason that Duffy et al. [5] and Travers et al. [7] advocated for a much broader understanding of complex illegal hunting and its drivers in order to design effective interventions. This justifies the need for conceptual views that provide a broader understanding on illegal hunting and what motivates people to hunt illegally.

Previous studies by Milner-Gulland and Leader-Williams [8], Hofer et al. [9], Damania et al. [10] and Keane et al. [11] developed models of relationships between illegal hunting and costs, benefits, sanctions, rewards and incentives. These models are mostly depicted in monetary form for benefits and highlighted law enforcement efforts (regulation) as a cost to illegal hunting. However, as observed by von Essen et al. [4], there are other non-monetary factors such as socio-political and normative values and beliefs that can influence hunters' illegal hunting behaviour. The economic models may not be robust enough to effectively represent the reality of the illegal hunting phenomenon. Therefore, despite providing some knowledge on the dynamics of illegal hunting, these models have application inadequacies in identifying relevant research variables and designing effective intervention measures against illegal hunting. Recently, Carter et al. [12] developed a conceptual framework for understanding illegal killing of large carnivores, which includes socio-economic, ecological and psychological factors and illustrates the complexity of illegal hunting. However, the study is focused on large carnivores which have specific illegal hunting risks and influences on the motivation to hunt illegally that might be different for other taxa. Thus, the conceptual framework by Carter et al. [12] requires validation of its applicability to wider taxa.

The literature on the illegal hunting phenomenon has emphasised instrumental economic theories despite other available perspectives, such as psychological and social-political, which also influence illegal hunting behaviour [4]. However, the significance of the psychological perspective in influencing illegal hunting behaviour is depicted in the Theory of Planned Behaviour, which holds that beliefs (attitude, subjective norms and perceived control) are determinants of both intentions and behaviour, with behavioural intentions being the most proximal determinant of social behaviour [13,14]. In relation to illegal hunting, the implication of the Theory of Planned Behaviour is that beliefs, norms and values towards illegal hunting determine the intention to hunt illegally and the illegal hunting behaviour. As a theoretical framework, the Theory of Planned Behaviour is therefore important to this study in facilitating the building of concepts on drivers of illegal hunting and in conceptualising how drivers of illegal hunting influence local hunters into illegal hunting behaviour. Considering that local hunters and the natural system and environment are linked and interdependent, the Socio-Ecological System (SES), as proposed by Ostrom [15], is adapted as this study's conceptual framework. Based on this conceptual framework, the local hunters are actors who are influenced by factors such as drivers of illegal hunting and others, and manifest illegal hunting behaviour within the system. Within the SES, the Theory of Planned Behaviour provides a theoretical underpinning for how drivers of illegal hunting influence local hunters' illegal hunting behaviour.

Notwithstanding a few studies on illegal hunting in Africa [1], studies in eastern, central, southern and western Africa that have identified drivers of illegal hunting provide some valuable information that can enhance cohesive conceptual understanding on the persistence of poaching on the continent. African countries may easily relate to one another in regard to the illegal hunting phenomenon because they have shared historical, socio-economic and political contexts. Therefore, the study uses a scoping review approach to provide an overview of the available evidence of what drives local hunters to hunt illegally in Africa based on the lived experiences of local hunters and not on perceptions from non-hunters. This study aims at developing concepts for drivers of illegal hunting and how they influence local hunters' illegal hunting behaviour in or adjacent to African protected areas. The study investigates the research question: what conceptual views can describe how drivers of illegal hunting influence local hunters' behaviours and the

persistence of illegal hunting in or adjacent to protected areas? The review-based conceptual framework of drivers of illegal hunting may provide a basis for contributing to a broader understanding of drivers of illegal hunting in Africa and help in identifying relevant elements for designing effective intervention measures that ensure sustainable wildlife conservation. Based on our knowledge, this is the first time drivers of illegal hunting by local hunters are conceptualised as underlying, proximate and thematic drivers of illegal hunting which are developed into a conceptual framework and used to explain the persistent illegal hunting in Africa.

## 2. Materials and Methods

### 2.1. Study Area

This scoping review is based on the studies that were conducted in five African regions (eastern, northern, western, central and southern). The islands in the Atlantic and Indian Oceans, such as Cape Verde, Comoros, Madagascar, Mauritius, Réunion and Seychelles, are part of the study area. The study focal sites are areas in or adjacent to forest or wildlife protected areas where hunting of wildlife without a permit is considered illegal.

### 2.2. Review Protocol

Prior to the review process, the review protocol was set to guide the process. Firstly, the review objectives and questions are: to identify available evidence on what drives local hunters to hunt illegally and to identify concepts on drivers of illegal hunting and how they influence illegal hunting using available evidence in Africa. The sources of available evidence are peer-reviewed articles, such as journal research papers, PhD and master's theses and book chapters. The review questions include: what concepts does the available evidence provide on drivers of illegal hunting by local hunters, and what conceptual framework can be developed from the available evidence to depict how drivers of illegal hunting influence illegal hunting behaviour? Secondly, to address these review objectives and questions, the study uses online searching of databases on drivers of illegal hunting in Africa with Google Scholar and ScienceDirect. The protocol has a pre-determined search strategy for identifying articles in the databases and criteria for inclusion and exclusion of identified articles (see below for details). Thirdly, the protocol addresses the extraction and presentation of data. Relevant data from included articles are identified during review and indicated in respective rows for each item in the table (see Supplementary Materials: Table S1). Simple frequency calculations are done for each identified item and the summarised data are presented in the results table. Fourthly, in fostering transparency, the search and identification of articles and summarising of data are done by the first author and the co-authors verify and approve the process and results.

### 2.3. Search Strategy

The Google Scholar and ScienceDirect database search engines were used to search for relevant studies on the drivers of illegal hunting by local hunters in Africa. The database was searched using phrases or words such as "bushmeat hunting", "drivers of illegal hunting", "hunters", "illegal hunting", "illegal killing", "motivation for illegal hunting", "motivation for poaching" and "poaching". The identified publications were initially screened for relevance to the objective of this study.

### 2.4. Inclusion and Exclusion Criteria

The relevance of the identified publications was further screened by checking if they met the inclusion and exclusion criteria indicated in Table 1. The publications were included or excluded depending on whether they met or failed to meet the criteria, respectively. Sampling local hunters in the publications was a critical criterion because hunters or resource users may have different experiences of what motivates them to hunt illegally from perceptions of non-hunters [16].

**Table 1.** Criteria for inclusion and exclusion of articles in the scoping review process.

| Criteria | Included | Excluded | Justification for the Criteria Used |
|---|---|---|---|
| Date of publication | 2000 to 2021 | Before 2000 and beyond 2021. | For current perspectives on drivers of illegal hunting and to access increased publications during this period. |
| Language of publication | English | All languages that are not English. | Researchers' proficiency in English language and to ensure increased readability. |
| Location of study | Publication on African countries. | Publications on non-African countries. | To maintain specific relevance and scope of the review. |
| Article availability | Available full papers identified though Google Scholar and ScienceDirect. | Full papers not accessible. | To access full/entire research findings from papers. |
| Type of articles | Peer-reviewed research journal articles, book chapters and PhD/master's theses. | Articles that are not peer-reviewed. | To ensure quality and validity of findings. |
| Publication content | Papers with drivers/motivations/ reasons for engaging in illegal hunting. | Papers without drivers/motivations/ reasons for engaging in illegal hunting. | To be specific and focused on the scope of the review. |
| Sampling methodology | Sampling local hunters through direct observations, interviews, questionnaires and focus group discussions. | Sampling non-hunters only. | To identify drivers of illegal hunting from lived experiences and not from perceptions by non-hunters. |

### 2.5. Identification and Analysis of Drivers of Illegal Hunting

The included publications were reviewed to identify drivers of illegal hunting by local hunters. The number and frequency of identification of each driver of illegal hunting were recorded to indicate levels of pervasiveness of respective drivers of illegal hunting in Africa. The included publications were also qualitatively analysed to identify behavioural intentions by local hunters to hunt illegally. Behavioural intentions to hunting illegally were expressed beliefs (behavioural, normative and perceived control) towards illegal hunting based on the Theory of Planned Behaviour [13,14]. According to the Theory of Planned Behaviour, beliefs (attitude, subjective norms and perceived control) are determinants of both intentions and behaviour, with behavioural intentions being the most proximal determinant of social behaviour [13,14]. Therefore, responses by local hunters in questionnaires, interviews and group discussions, as reported or quoted in the included publications, that depicted behavioural, normative or perceived control beliefs were used to determine behavioural intention towards illegal hunting. Beliefs expressed as 'hunting is our birth right or cultural right', 'hunting is the only way to support my family', 'we have no other option apart from hunting' or 'we have other ways to outwit anti-poaching measures' were indicative of the behavioural intention to hunt illegally by local hunters. In this study, behavioural intentions to hunt illegally were considered drivers of illegal hunting based on the Theory of Planned Behaviour [17].

### 2.6. Proximate, Underlying and Thematic Drivers of Illegal Hunting

The identified drivers of illegal hunting from reviewed publications were categorised into proximate and underlying drivers. This categorisation follows the descriptions of proximate and underlying drivers adapted from those on tropical deforestation and conversion

of natural vegetation to agricultural land use in Africa by Geist and Lambin [18] and Jellason et al. [19], respectively. Based on these adaptations, we characterised proximate drivers of illegal hunting as any immediate desires, feelings, shortages or needs by humans at a local level that directly trigger them to hunt illegally. Similarly, we characterised underlying drivers as factors that underpin, enhance or enable proximate drivers and may also work at the local level or have an indirect influence from the national or global levels. Further, the identified drivers of illegal hunting were qualitatively analysed and categorised into thematic drivers. Thematic drivers were determined by considering similarities or related characteristics of both proximate and underlying drivers and assigning them appropriate respective thematic driver categories.

### 2.7. Conceptual Framework of Drivers of Illegal Hunting by Local Hunters

The conceptual framework of drivers of illegal hunting was developed to depict the process of how underlying and proximate drivers, working with other social and ecological influences and constraints, affect illegal hunting behaviour. The conceptual framework was informed by and adapted from the Socio-Ecological System (SES) framework by Ostrom [15]. The relevance of SES to the development of a conceptual framework of drivers of illegal hunting by local hunters is that it considers human and natural systems as being linked and interdependent. Therefore, illegal hunting behaviour manifests in a natural system with complex, linked and interdependent components. The development of the conceptual framework of how drivers of illegal hunting affect illegal hunting behaviour was based on the Theory of Planned Behaviour [17].

## 3. Results

### 3.1. Search Results

The scoping review process identified 30 publications that were included for review from the initial 1014 articles identified using Google Scholar ($n$ = 995) and ScienceDirect ($n$ = 19) (see Figure 1). A total of 948 articles were excluded from 997 retrieved duplicate free articles, owing to the studies not being conducted in Africa, inaccessible full papers, and not addressing drivers or motivations for illegal hunting. A total of 19 full-text articles were excluded from 49 full-text articles which were assessed for eligibility for not sampling local hunters. The 30 studies which were included for review were conducted in 17 countries, with 13 studies from eastern, 10 from southern, 4 from western and 3 from central regions of Africa (Supplementary Materials: Table S1).

### 3.2. Identified Drivers of Illegal Hunting

A total of 17 drivers of illegal hunting were identified in the publications that were included for review (see Table 2). The need to generate income/no income source was the most frequently identified driver of illegal hunting by 26 studies (86.7%), followed by the need/preference for bushmeat consumption, identified by 25 studies (83.3%). The third-ranking drivers of illegal hunting were cultural needs/rights and preventative/retaliatory killing, both identified by 11 studies (36.7%). Poverty and weak/inadequate law enforcement were identified by 6 (20.0%) and 4 (13.3%) studies and ranked seventh and eighth among drivers of illegal hunting, respectively. Notably, defiance/protest as a driver of illegal hunting was identified by studies conducted in southern African regions only (see Supplementary Materials: Table S1).

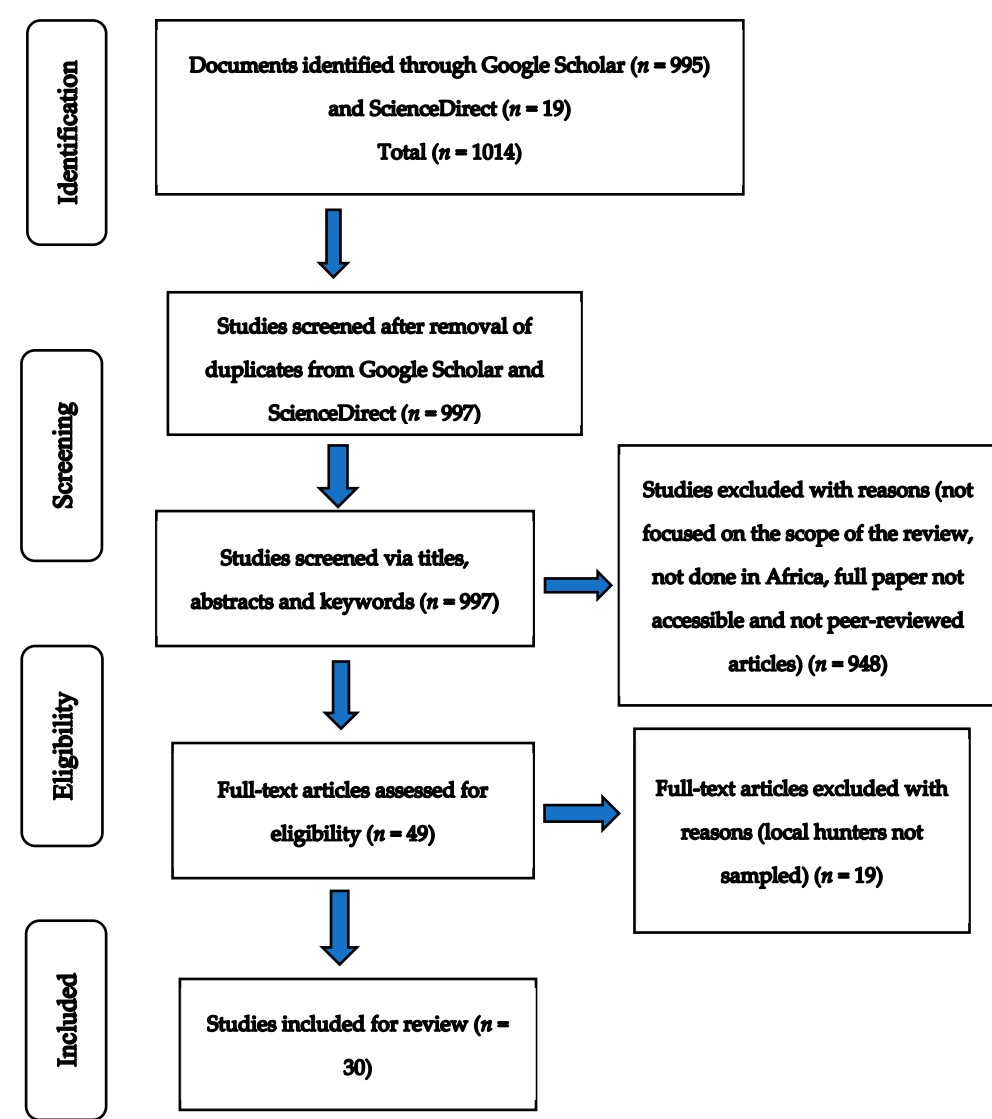

**Figure 1.** Flowchart indicating number of publications included and excluded during the scoping review process (based on Moher et al. [20]).

**Table 2.** Identified, proximate, underlying and thematic drivers of illegal hunting of wildlife derived from scoping review of publications (published from 2000 to 2021) that surveyed experiences of local hunters living in or adjacent to African protected areas.

| Identified Drivers of Illegal Hunting | No. of Publications with Identified Drivers | Frequency of Identification of Drivers by Publications | Classification of Driver (Proximate or Underlying) | Thematic Drivers |
|---|---|---|---|---|
| Need to generate income/ no income source | 26 | 86.7% | Proximate | Need for survival and sustaining livelihoods |
| Need/preference for bushmeat consumption | 25 | 83.3% | Proximate | Need for survival and sustaining livelihoods |
| Cultural needs/rights | 11 | 36.7% | Proximate | Cultural needs/significance |

**Table 2.** *Cont.*

| Identified Drivers of Illegal Hunting | No. of Publications with Identified Drivers | Frequency of Identification of Drivers by Publications | Classification of Driver (Proximate or Underlying) | Thematic Drivers |
|---|---|---|---|---|
| Preventative/retaliatory killing of wildlife | 11 | 36.7% | Proximate | Human–wildlife conflict |
| Behavioural intention to hunt illegally | 9 | 30.0% | Proximate | Behavioural intention to hunt illegally |
| Lack of employment/ livelihoods | 8 | 26.7% | Proximate | Need for survival and sustaining livelihoods |
| Shortage/expensive/lack of protein source | 7 | 23.3% | Proximate | Need for survival and sustaining livelihoods |
| Poverty | 6 | 20.0% | Underlying | Need for survival and sustaining livelihoods |
| Weak/inadequate law enforcement | 4 | 13.3% | Underlying | Inadequate legislation/ enforcement |
| Defiance/protest | 3 | 10.0% | Proximate | Defiance/protest |
| Political instability/armed warfare | 3 | 10.0% | Underlying | Political/armed conflicts |
| Demand for wildlife products | 2 | 6.7% | Underlying | Market demand for wildlife products |
| Recreational/sports needs | 2 | 6.7% | Proximate | Recreational needs |
| Population influx/increase | 2 | 6.7% | Underlying | Demographic growth |
| Need for traditional medicine | 2 | 6.7% | Proximate | Need for survival and sustaining livelihoods |
| Social status identity | 1 | 3.3% | Proximate | Cultural needs/significance |
| Need for skins and bones | 1 | 3.3% | Proximate | Cultural needs/significance |

### 3.3. Proximate, Underlying and Thematic Drivers of Illegal Hunting

Among the identified drivers of illegal hunting, 12 (70.6%) were categorised as proximate drivers and 5 (29.4%) as underlying drivers. The first seven most frequently identified drivers of illegal hunting were proximate drivers and included the need for income generation (86.7%), the need for bushmeat consumption (83.3%), cultural needs/rights (36.7%), preventative/retaliatory killing of wildlife (36.7%), behavioural intention to hunt illegally (30.0%), lack of employment/livelihood (26.7%) and shortage/expensive/lack of protein source (23.3%). The underlying drivers were among the last 10 least frequently identified drivers of illegal hunting. Poverty (20.0%) and weak/inadequate law enforcement (13.3%) were the most frequently identified drivers among the underlying drivers.

The identified drivers of illegal hunting were further categorised into 10 thematic drivers (Table 2). Six identified drivers of illegal hunting were thematically categorised as need for survival and sustaining livelihoods, and five of these were ranked among the eight most frequently identified drivers by the included publications. The six identified drivers that contributed to the thematic driver of the need for survival and sustaining livelihoods were basically socio-economic drivers and included the need to generate income, need to consume bushmeat, lack of employment/livelihoods, shortage/expensive/lack of protein source, poverty and the need for traditional medicine. The second-ranking thematic driver of illegal hunting was cultural needs/significance and included cultural needs/rights, social status identity and the need for skins and bones.

The human–wildlife conflict was the third most identified thematic category of drivers of illegal hunting, with preventative/retaliatory killing of wildlife being a contributing driver. Next, after behavioural intention to hunt illegally, is the fifth-ranked thematic driver of illegal hunting, categorised as inadequate legislation/enforcement.

### 3.4. Conceptual Framework of Drivers of Illegal Hunting by Local Hunters

The conceptual framework was developed to show how underlying, proximate and most proximate drivers (behavioural intentions) influence illegal hunting behaviour (Figure 2). The drivers of illegal hunting behaviour are influenced and constrained by socio-ecological factors within the SES.

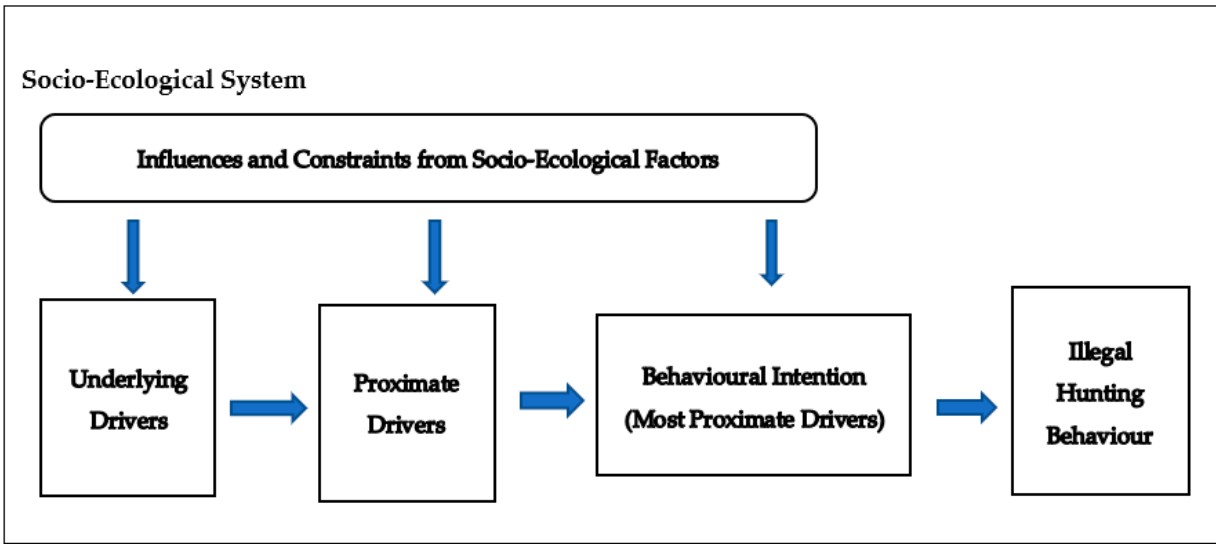

**Figure 2.** Conceptual framework of how underlying and proximate drivers and behavioural intentions influence illegal hunting behaviour of local hunters living in or adjacent to protected areas in Africa (based on Ostrom [15]).

## 4. Discussion

### 4.1. Identified Drivers of Illegal Hunting

This study shows that the needs to generate income (or lacking income sources) (86.7%) and consume bushmeat (83.3%) are the two most prevalent drivers of illegal hunting identified by studies among local hunters in Africa. Since illegal hunting is a wildlife crime that is considered an integral part of the illegal wildlife trade [21], it can be argued, therefore, that the illegal wildlife trade has thrived among local people in Africa primarily because local hunters are mostly driven to hunt illegally by the needs to generate income and consume bushmeat. Despite being less prevalently identified in publications, the other drivers of illegal hunting, such as cultural needs (36.7%), lack of employment (26.7%), lack of protein sources (23.3%), poverty (20.0%), demand for wildlife products (6.7%) and need for traditional medicine (6.7%), may play complementary roles to the two most prevalent drivers of the illegal wildlife trade among local hunters in Africa.

The study identified behavioural intention to hunt illegally (in 30% of included publications) as a driver of illegal hunting by analysing further the responses from hunters in the included publications. The survey and identification of behavioural intention to hunt illegally as one of the drivers of illegal hunting were not planned in the included publications for review. Other studies on drivers of illegal hunting have also not identified behavioural intention to hunt illegally as a driver of illegal hunting. Therefore, the identification and inclusion of behavioural intention as one of the drivers of illegal hunting is a novelty that may have implications on enhancing understanding on illegal hunting behaviour and potential intervention measures. The Theory of Planned Behaviour [14,17] provides support and a basis for adopting behavioural intention to hunt illegally as a driver of illegal hunting behaviour. The theory is particularly relevant, as it has been used in investigating potential predictors of illegal hunting and as a framework for assessing intervention measures against illegal hunting [22,23].

The defiance or protest against injustice and illegitimate authorities or rules as a driver of illegal hunting appear to be associated with the southern African region. This may be attributed to the historical background in southern Africa, where local people were racially discriminated against, dispossessed of land and disenfranchised from accessing land, wildlife and forest resources by colonial and militarised conservation authorities [24]. Local people hunt wildlife in defiance of rules and authorities and in protest against any perceived injustices that deny them to use resources that are considered a birthright [25].

*4.2. Proximate, Underlying and Thematic Drivers of Illegal Hunting*

The study here conceptualised the need for survival and sustaining livelihoods as the key driver of illegal hunting in Africa. Studies in Africa have provided empirical evidence that illegal hunting is a strategy employed by local hunters for survival and sustaining livelihoods [26–30]. Accordingly, this study has not conceptualised poverty as the key driver of illegal hunting in Africa, but as one of the underlying drivers which contributes to the major thematic driver of illegal hunting—the need for survival and sustaining livelihoods. Whereas some studies have indicated that poverty is a key driver of illegal hunting [31,32], others have shown that there is no evidence that poverty is the major driver of illegal hunting in some areas in Africa [7,33]. This implies that the role of poverty as a driver of illegal hunting varies across Africa. The contribution of poverty to the key thematic driver of poaching may be major or less significant, depending on the socio-economic situation of a study area. Therefore, conceptualising poverty as one of the contributing drivers to the key thematic driver of illegal hunting (need for survival and sustaining livelihoods) provides a unifying framework for both views.

In this study, cultural needs/significance was the second most identified thematic driver of illegal hunting among local hunters. A recent global systematic review by Lavadinović et al. [34] also identified socio-cultural influence as the third most prevalent identified driver of poaching. The corroborative findings of studies by Lavadinović et al. [34] and this study suggest that cultural factors (values, beliefs and norms towards wildlife) may have profound effects on hunter's behaviour towards wildlife species and their habitats and on societal responses to illegal hunting [35,36]. However, data on and understanding of socio-cultural values, beliefs and norms towards wildlife, bushmeat and the environment have been low [37]. It is only recently that recognition of the importance of considering how socio-cultural contexts influence illegal hunting and strategies for curbing the poaching problem is being made [36,38].

This study found that human–wildlife conflict was the third most identified thematic driver of illegal hunting in Africa, and thereby confirmed its importance in threatening conservation in Africa [39]. Human–wildlife conflict has direct major consequences for humans in rural people's development, income, health, food security and livelihoods [40], making it appropriate to be categorised under the thematic driver—the need for survival and sustaining livelihoods. This may additionally augment the need for survival and sustaining livelihoods as the key driver of illegal hunting in Africa. However, because human–wildlife conflict also has direct consequences on the conservation of individual animals, species and the broader ecosystem and biodiversity in African landscapes [39,40], it has been categorised under a separate thematic driver of illegal hunting for the need for survival and sustaining livelihoods. Considering human–wildlife conflict as a separate category, which is the third most identified thematic driver of illegal hunting, highlights it as a major African concern that has consequences on development, livelihoods and conservation.

A few studies (4, 13.3% frequency) identified weak/inadequate law enforcement as a driver of illegal hunting by local hunters, implying that it may not be a prevalent driver of illegal hunting to local hunters in Africa. Weak/inadequate law enforcement ranked as the eighth most identified driver of illegal hunting by local hunters. The low frequency of identification of weak/inadequate law enforcement as a driver of poaching by this study is surprising considering that other studies show that law enforcement is the most prioritised in terms of investments and the most studied among intervention measures against illegal

hunting in Africa [1,41,42]. Ideally, the most prioritised intervention efforts in terms of investments should be targeted at addressing the most prevalent driver of illegal hunting by local hunters. However, the less prevalent driver of illegal hunting (weak/inadequate law enforcement) reported here is apparently targeted by highly prioritised efforts. In the identified studies, few local hunters indicated weak/inadequate law enforcement as a motivation for hunting illegally, probably because, as an underlying driver, it is indirect and less relevant to local hunters in influencing them to hunt illegally. It is similarly surprising that local hunters in the included publications did not indicate lack of/inadequate involvement of local communities in the management of wildlife as a driver of their poaching behaviour. However, it has been argued that involvement of local communities in wildlife management should be prioritised as a potentially effective intervention for addressing illegal hunting [3,43,44]. The probable explanation for this apparent conflicting situation again is that lack of/inadequate involvement in the management of wildlife is an underlying driver which may not directly influence local hunters to hunt illegally. This underscores the importance of understanding what motivates local hunters to hunt illegally, which might be different from perceptions of non-hunters located in or adjacent to African wildlife protected areas [16].

### 4.3. Conceptual Framework of Drivers of Illegal Hunting by Local Hunters

The proposed conceptual framework provides a relevant and enhanced understanding of illegal hunting behaviour of local hunters who live in or adjacent to African protected areas. Firstly, the novel inclusion of a behavioural intention to hunt illegally component as driver of illegal hunting behaviour may provide an expanded understanding of drivers of illegal hunting behaviour. Previous studies had not considered behavioural intention to hunt illegally as a driver of poaching, and thus it has had no specific intervention measures to address it. The Theory of Planned Behaviour [17] posits that behavioural intention is the most proximate determinant of an illegal behaviour, which therefore mediates proximate drivers and the illegal hunting behaviour in the conceptual framework. Behavioural intention is a function of beliefs which result from psychological aspects such as attitudes, subjective norms and perceived control. Based on the Theory of Planned Behaviour, the most proximate psychological drivers of illegal hunting in the proposed conceptual framework are probably the most critical drivers that energise other various drivers of illegal hunting to influence local hunters into illegal hunting behaviour. Secondly, the proposed conceptual framework may not be complicated, but represents complex linkages and interdependent components from underlying, proximate and most proximate drivers and socio-ecological factors. The socio-ecological factors and drivers of illegal hunting may be social, political, economic, psychological and ecological in nature. The linkages and interdependence of components in the framework underscores the appropriateness of basing the construction of the proposed framework on SES. The local hunters, as actors in the framework, are influenced by factors such as underlying and proximate drivers of illegal hunting, other socio-ecological factors and behavioural intentions to manifest illegal hunting behaviour. Therefore, this proposed conceptual framework may provide further understanding of how drivers of illegal hunting influence illegal hunting behaviour and what components should be targeted when tackling illegal hunting.

### 4.4. Implications on Interventions and Persistence of Illegal Hunting in Africa

The findings of this study have practical implications on the effectiveness of intervention measures and the persistence of illegal hunting in Africa. Firstly, the prioritised and increased efforts of law enforcement [1,41,42] may not address the main driver of illegal hunting by local hunters in Africa. Whereas the key driver of illegal hunting by local hunters is the need for survival and sustaining livelihoods, the main intervention measure is instead law enforcement. Local hunters who are motivated by the need to survive and sustain livelihoods may not be deterred by law enforcement but continue to hunt illegally by changing hunting tactics to those that are not easily detectable by law

enforcement workers [45,46]. This is because law enforcement is used as a measure for dealing with illegal hunting activities and not the drivers. Further, Milner-Gulland and Leader-Williams [8] found that very high levels of enforcement deterred outsider-organised hunters and not local hunters and reported that local hunters responded positively to community schemes that provided employment for addressing poaching. The positive response to community schemes is because it addressed the main driver of illegal hunting among local hunters. Therefore, using law enforcement to address illegal hunting that is mainly driven by the need to survive and sustain livelihoods among local hunters is a mismatched intervention measure. Under these circumstances, illegal hunting by local hunters may persist decimating wildlife in African protected areas.

Secondly, behavioural intention to hunt illegally has not been considered one of the drivers of poaching, and therefore no specific intervention measures have been designed to address it. Behavioural intention is a function of beliefs (behavioural or attitude towards hunting, normative and perceived control) and are crucial in influencing illegal hunting behaviour. People living in the same location and influenced by the same underlying and proximate drivers may exhibit opposite behaviours (some hunting illegally and others not hunting) owing to differences in their behavioural intentions towards illegal hunting. Therefore, illegal hunting may have persisted in Africa due to lacking specific intervention measures for addressing behavioural intentions to hunt illegally.

Thirdly, each identified underlying, proximate and most proximate (behavioural intention) driver provides the contextual bases for identifying relevant elements and designing specific intervention measures against illegal hunting motivations. Each identified driver of illegal hunting should be addressed by specific intervention measures to ensure poaching is effectively tackled. Since prevalence levels of the drivers of illegal hunting are different, as observed in this study, levels of efforts for intervention measures should be distributed accordingly. However, the efforts and investments for addressing the most prevalent drivers of illegal hunting, such as the need for survival and sustaining livelihoods, have been left disproportionately less than those of law enforcement [41,42]. As a result, the main driver of illegal hunting among local hunters may not be effectively addressed and this may consequently facilitate the persistence of illegal hunting in Africa.

Fourthly, despite being the second most identified thematic driver of illegal hunting among local hunters in Africa, cultural needs/significance, and particularly the cultural constructions that shape values, attitudes, beliefs and norms towards wildlife in respective contexts, are inadequately known and understood [37]. Inadequate data on and understanding of socio-cultural factors are major constraints in designing specific intervention measures for addressing or mitigating the cultural needs/significance as a driver of poaching. As such, intervention measures that are based on inadequate understanding are likely to fail in addressing illegal hunting that is driven by the cultural needs/significance among local hunters and may thereby contribute to persistence of illegal hunting in Africa.

Fifthly, the third most identified thematic driver of illegal hunting in Africa, the human–wildlife conflict, is escalating globally because of competition for space and resources, such as water and food, by wildlife and growing human populations and expanded cultivation and livestock husbandry, which is influencing increased illegal preventative and retaliatory killings due to crop and livestock depredation by wildlife [39,40,47]. However, efforts to address human–wildlife conflict have been failing, as the level of the solutions does not match the level of the problem, and they are usually not applied to scale and holistically [40]. The escalating human–wildlife conflict is probably another reason contributing to persistence of illegal hunting in Africa.

Based on the foregoing, we postulate that the persistent (and prevalent) illegal hunting (and, implicitly, the illegal wildlife trade) by local hunters in or adjacent to African wildlife protected areas may be associated mainly with two factors: the first factor is the prevalence of drivers of illegal hunting that are related to the need for survival and sustaining livelihoods among local hunters. These drivers of illegal hunting include the need to generate income, need for bushmeat consumption, lack of employment, poverty and the

need for traditional medicine. As discussed earlier, the human–wildlife conflict may also appropriately be included among drivers that relate to the need for survival and sustaining livelihoods among local hunters. Secondly, the persistent illegal hunting by local hunters in Africa is also probably associated with the prevalence of unaddressed or ineffectively addressed drivers of illegal hunting. The unaddressed or ineffectively addressed drivers of poaching in this regard relate mostly to the need for survival and sustaining livelihoods among local hunters.

*4.5. Limitations of the Study*

The study did not consider hunters who were not living in nor adjacent to protected areas and perceptions of local people or stakeholders who are non-hunters on what drives local hunters to hunt illegally in areas. Therefore, the study may not provide further understanding of whether perceptions of non-hunters on drivers of illegal hunting are different to drivers indicated by local hunters. Consequently, the study may also not determine whether surveying perceptions from non-hunters on drivers of illegal hunting is valid for use in designing intervention measures. Another limitation is the exclusion criterion for articles not published in the English language. This could have biased results because there are countries in Africa with official languages that are not English. This could have been mitigated by using translators for articles published in other languages. However, it was assumed that numbers of articles published on the subject matter in non-English languages were very small and it was observed in this study that there were articles published in English from regions with non-English official languages. Furthermore, another limitation of this study is that searches on Google Scholar databases retrieved more articles than those on ScienceDirect databases; in particular, Google Scholar has limitations for use as a single database search source [48]. Using both Google Scholar and ScienceDirect would suffice for a scoping review which is an overview study where assessments for quality and effectiveness assessment may not be required.

*4.6. Future Directions*

We recommend that comparable studies that consider hunters that live farther away from protected areas and non-hunters who live in or adjacent to protected areas be conducted to determine drivers and perceptions, respectively. The studies would compare findings on drivers from these populations and determine if perceptions by non-hunters are different from the experiences of local and distant hunters. Site-specific studies may be conducted for testing, comparison and validation of the conceptual views reported in this study. We also recommend for studies to enhance understanding of the identification of the behavioural intention to hunt illegally, as the most proximate driver of illegal hunting, and of how the adoption of intervention measures for addressing it affects illegal hunting behaviour. Furthermore, we recommend that when designing intervention measures, it is critical to ensure that measures are not designed to deal with superficial illegal hunting activities, but the causes which are drivers of illegal hunting behaviour. Therefore, all drivers of illegal hunting (including behavioural intentions to hunt illegally) should be identified first and then specific intervention measures for each driver should be designed.

## 5. Conclusions

The problem of inadequate evidence-based information has led to ineffective and restricted choices of intervention measures for tackling illegal hunting that is persistently decimating wildlife populations in African protected areas. However, we believe this study has contributed to enhancing understanding of illegal hunting and what motivates people to hunt illegally. Firstly, the study identified behavioural intention to hunt illegally as one of the 17 drivers of illegal hunting by local hunters in Africa, which hitherto had not been considered a driver of illegal hunting by previous studies. Secondly, the study conceptualised identified drivers of illegal hunting as proximate, underlying and thematic. Consequently, the need for survival and sustaining livelihoods was conceptualised as the

main thematic driver of illegal hunting in Africa, a narrative that is different from the one that highlights poverty as a key driver. Thirdly, the conceptual framework suggested by this study represents how drivers of illegal hunting influence illegal hunting behaviour and provides a novel aspect that might enhance further understanding on illegal hunting by local hunters in Africa. The findings of this study may be helpful to researchers and conservationists in providing concepts, statistics and frameworks on drivers of illegal hunting for application in identifying relevant variables for designing research projects, intervention measures and strategies for tackling illegal hunting drivers and ensuring sustainability in wildlife conservation.

**Supplementary Materials:** The following supporting information can be downloaded at: https://www.mdpi.com/article/10.3390/su141811204/s1, Table S1: Summary of included publications with drivers of illegal hunting by local hunters in Africa identified using a scoping review of literature published from 2000 to 2021. References [7,24,25,27,28,30,32,33,43,49–69] are cited in the Supplementary Materials.

**Author Contributions:** P.Z.—Conceptualisation of the study; design of methodology; conducting literature search; data analysis; interpretation of results, writing, reviewing and editing the draft manuscript. F.K.K.—Providing advice on conceptualisation of the study, supervision of the study, interpretation of results and making corrections to the draft. J.M.—Providing advice on conceptualisation of the study, scholarly supervision of the study, interpretation of results and making corrections to the draft. V.R.N.—Providing advice on conceptualisation of the study, interpretation of results, making comments and corrections to the draft. All authors have read and agreed to the published version of the manuscript.

**Funding:** This research received no external funding.

**Institutional Review Board Statement:** This is not applicable, as this study is a review of already peer-reviewed publications, and the study does not involve contact with animal and human targets.

**Informed Consent Statement:** This is not applicable, as this study is a review of already peer-reviewed publications and the study does not involve contact with animal and human targets.

**Data Availability Statement:** Data are contained within the article and Supplementary Material: Table S1.

**Acknowledgments:** This study is part of a PhD study by the first author, and the academic staff in the School of Postgraduate Studies at the University of Lusaka provided valuable advice and guidance.

**Conflicts of Interest:** The authors declare no conflict of interest.

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
