# Peer review of "Conceptualising Drivers of Illegal Hunting by Local Hunters Living in or Adjacent to African Protected Areas: A Scoping Review"

_sustainability, doi:10.3390/su141811204_

Round 1

Reviewer 1 Report

I commend the authors on a well-written and insightful paper on a very timely topic! I have included comments within the attached PDF, with specific feedback relevant to the Intro and Discussion.

Author Response

RESPONSES TO REVIEWER 1

Point 1: Can we get just a bit more explanation on the Duffy et al/Traverse et al calls for broader understandings?

Response 1: More explanation has been added to the paragraph in three sentences as reasons why Duffy et al/Traverse et al called for broader understanding. The added sentences appear prior to the call by Duffy et al/Traverse et al. in the same paragraph.

Point 2: Great point

Response 2: Thank you.

Point 3: If possible, I think a full of key words would be helpful. Were any key words related to location (eg Africa, central Africa, etc)?

Response 3: There were no key words related to location, such as including 'in Africa' for each search phrase or word. However, location was used as one of the items in the criteria for inclusion or exclusion of articles and was done by checking the titles and study area sections.

Point 4: This may have biased results given the study area, correct? I'd address this as a limitation.

Response 4: That is correct, language of publication may have biased results because there are countries in Africa with non-English as official language. It has been dealt with under limitation of the study section as suggested.

Point 5: For each of these criteria (specifically English language, article availability, type of article, and sampling), it would be helpful to know how many were initially pulled and then how many were excluded based on the criteria.

Edit: I see this is somewhat addressed in Fig 1, but a more comprehensive run down would be helpful.

Response 5: For English language it was not possible to know what was pulled out because English language was used as a filter by Google Scholar search engine which only allowed articles in English language to be retrieved. For sampling local hunters, inclusion or exclusion was done after reviewing full texts as a last phase. So what was pulled is represented by full text articles checked for sampling local hunters. This criterion is considered critical to the study and the pulled and excluded numbers are thus reported and highlighted as suggested. The article availability and type of articles were pooled and numbers are indicated in the Figure 1 and described in the search results section.

Point 6: Does this mean that modeling papers or others that did not specifically sample hunters were excluded?

Response 6: Yes, modelling papers and others that did not sample hunters or use data from hunters were excluded. This is because the interest of the study is in providing evidence from lived experiences of hunters and not perceptions from non-hunters.

Point 7: I think TPB should be mentioned in the Intro (and perhaps explored in this context) if it's a major part of this study. Then I'd dive into it within a "Theoretical Model" section or something similar. Then you could have this section (perhaps shortened) to explain how this study applies TPB.

Response 7: Thank you for this helpful observation. A paragraph on TPB and how it applies to the study has been included to the Introduction section as suggested. It has also included SES and highlighted both (TPB and SES) as theoretical and conceptual perspectives of the study respectively.

Point 8: Could you perhaps give an example or more clarification about how proximate and underlying drivers were then consolidated thematically? Why consolidate them?

Response 8: Thank you for your observation. Actually, proximate and underlying drivers were not consolidated. They were qualitatively assessed for similarities and related characteristics and then assigned appropriate respective thematic driver categories. The sentence was accordingly edited for clarity.

Point 9: I'd also include this in the Intro/mention the use of SES in an illegal hunting context.

Response 9: As responded in response 7 above, SES and its use in an illegal hunting context has been included in the introduction.

Point 10: were identified?

Response 10: The sentence has been edited to include what was omitted.

Point 11: Given the importance of TPB and SES within the Methods section, I think more can be discussed here. For example, TPB includes attitudes, subjective norms, and perceived control that result in behavioral intention, perhaps those factors can be further discussed here.

Response 11: TPB and SES are briefly discussed under discussion section 4.3 as suggested. 

Point 12: yes!

Response 12: Thank you for affirmation

Please see the attached revised manuscript with track changes. Please note that the revised manuscript article has responses for both Review 1 and Review 2.

Reviewer 2 Report

This is a very well-written and summarized review of the drivers of illegal hunting for local African hunters. There are almost no flaws in the introduction and discussion sections. However, my main point of contention is whether or not this can be classified as a systematic review. The manuscript does not address all the points in the PRISMA checklist to be considered a systematic review. Moreover, the use of Google Scholar as the sole database for a systematic review is also considered less than ideal (Gusenbauer, M., & Haddaway, N. R. (2020). Which academic search systems are suitable for systematic reviews or meta‐analyses? Evaluating retrieval qualities of Google Scholar, PubMed, and 26 other resources. Research synthesis methods11(2), 181-217.; Giustini, D., & Boulos, M. N. K. (2013). Google Scholar is not enough to be used alone for systematic reviews. Online journal of public health informatics5(2), 214.). Therefore, I cannot recommend the publication of the manuscript in its current form, as long as it refers to itself as a systematic review. Please consider adhering to the PRISMA guidelines and including other databases as part of the literature search. Alternatively, please reclassify it as another type of review, e.g. critical review.

Author Response

RESPONSES TO REVIEWER 2

Point 1: This is a very well-written and summarized review of the drivers of illegal hunting for local African hunters. There are almost no flaws in the introduction and discussion sections. However, my main point of contention is whether or not this can be classified as a systematic review. The manuscript does not address all the points in the PRISMA checklist to be considered a systematic review. Moreover, the use of Google Scholar as the sole database for a systematic review is also considered less than ideal (Gusenbauer, M., & Haddaway, N. R. (2020). Which academic search systems are suitable for systematic reviews or meta‐analyses? Evaluating retrieval qualities of Google Scholar, PubMed, and 26 other resources. Research synthesis methods11(2), 181-217.; Giustini, D., & Boulos, M. N. K. (2013). Google Scholar is not enough to be used alone for systematic reviews. Online journal of public health informatics5(2), 214.). Therefore, I cannot recommend the publication of the manuscript in its current form, as long as it refers to itself as a systematic review. Please consider adhering to the PRISMA guidelines and including other databases as part of the literature search. Alternatively, please reclassify it as another type of review, e.g. critical review.

Response 1: Thank you for helpful comments. Considering the guidance provided, the review article was NOT appropriately titled as systematic review because most PRISMA guidelines were adhered to and the use of only Google Scholar for database search. Furthermore, the study did not involve quality appraisal or feasibility or effectiveness.

The review is changed to Scoping Review because the study addressed an overview of available evidence on illegal hunting aimed at building concepts from available evidence instead of detailed and specific review. The article has been edited accordingly  (title, abstract, introduction, methods, results and discussion). Further, another database (ScienceDirect) was included in the study.

A section on Review protocol was included under Methods. The protocol addresses the objectives, questions, sources of evidence, how data is extracted and presented and transparency according to Scoping review guidelines.

Please see the attached revised manuscript with track changes. Please note that the revised manuscript article has responses for both Review 1 and Review 2.

Round 2

Reviewer 1 Report

I thank the authors for the edits they've made. I think this manuscript is stronger now.

Reviewer 2 Report

I am satisfied with the changes made in the revised manuscript.